# Mechanism of β-Catenin in Pulmonary Fibrosis Following SARS-CoV-2 Infection

**DOI:** 10.3390/cells14060394

**Published:** 2025-03-07

**Authors:** Min Jiang, Jiaqi Hou, Qianqian Chai, Shihao Yin, Qian Liu

**Affiliations:** Department of Forensic Medicine, Tongji Medical College, Huazhong University of Science and Technology, Wuhan 430074, China; m202275690@hust.edu.cn (M.J.); m202375900@hust.edu.cn (J.H.); m202375899@hust.edu.cn (Q.C.); 2021612245@hust.edu.cn (S.Y.)

**Keywords:** β-catenin, pulmonary fibrosis, SARS-CoV-2, transcriptomics, WNT signaling

## Abstract

Pulmonary fibrosis due to severe acute respiratory syndrome coronavirus 2 (SARS-CoV-2) infection is the leading cause of death in patients with COVID-19. β-catenin, a key molecule in the Wnt/β-catenin signaling pathway, has been shown to be involved in the development of pulmonary fibrosis (e.g., idiopathic pulmonary fibrosis, silicosis). In this study, we developed a SARS-CoV-2-infected A549-hACE2 cell model to evaluate the efficacy of the A549-hACE2 monoclonal cell line against SARS-CoV-2 infection. The A549-hACE2 cells were then subjected to either knockdown or overexpression of the effector β-catenin, and the modified cells were subsequently infected with SARS-CoV-2. Additionally, we employed transcriptomics and raw letter analysis approaches to investigate other potential effects of β-catenin on SARS-CoV-2 infection. We successfully established a model of cellular fibrosis induced by SARS-CoV-2 infection in lung-derived cells. This model can be utilized to investigate the molecular biological mechanisms and cellular signaling pathways associated with virus-induced lung fibrosis. The results of our mechanistic studies indicate that β-catenin plays a significant role in lung fibrosis resulting from SARS-CoV-2 infection. Furthermore, the inhibition of β-catenin mitigated the accumulation of mesenchymal stroma in A549-hACE2 cells. Additionally, β-catenin knockdown was found to facilitate multi-pathway crosstalk following SARS-CoV-2 infection. The fact that β-catenin overexpression did not exacerbate cellular fibrosis may be attributed to the activation of PPP2R2B.

## 1. Introduction

Severe acute respiratory syndrome coronavirus 2 (SARS-CoV-2) is a single-stranded, positive-sense RNA virus that has infected 777 million people and caused 7.1 million deaths globally as of December 2024 (World Health Organization, WHO). While numerous reports highlight multiorgan damage and failure caused by SARS-CoV-2 infection, respiratory symptoms and lesions continue to be the most severe manifestations. SARS-CoV-2 spike (S) proteins promote host cell invasion by binding to the C-terminal structural domain of the human angiotensin-converting enzyme 2 (hACE2) complex, which facilitates the formation of a crystalline structure and subsequent membrane fusion [1,2]. The SARS-CoV-2 genome has evolved and mutated over time. Currently, the WHO has designated the following as variants of concern: Alpha, Beta, Gamma, Delta, and Omicron [3]. The Alpha variant results in more severe long-term sequelae, such as lung damage, compared to the Omicron variant [4].

Patients who have succumbed to infections caused by the SARS-CoV-2 Alpha strain exhibit varying degrees of pulmonary fibrosis (PF) [5]. PF occurs when epithelial cells lose their epithelial markers, migrate to the lamina propria, and acquire mesenchymal markers, a process known as epithelial–mesenchymal transition [6]. Most cases of PF arise from previous acute lung inflammation. If these inflammatory reactions are not resolved promptly, they can result in the deposition of fibrous tissue in the lungs, which may subsequently lead to the development of PF [7]. During the initiation phase of PF, stress and immune responses trigger the activation of multiple pro-inflammatory pathways. This is followed by a proliferative phase in which fibroblasts undergo differentiation and proliferation. Finally, the modification phase involves a reorganization of the extracellular matrix (ECM), comprising immune cells and fibroblasts [8]. The ECM consists of a variety of proteins, among which fibronectin (FN) is a significant glycoprotein. FN is one of the first ECM proteins to be assembled during the initial stages of tissue development and wound healing [9]. Type III collagen (COL3A1), an ECM protein, is synthesized by cells as a pre-procollagen [10]. α-smooth muscle actin (α-SMA) is a well-characterized protein that serves as a marker for activated fibroblasts in various tissues and organs, including the lung [11]. Vimentin, a member of the intermediate filament family of proteins, is essential for epithelial–mesenchymal transition [12]. In addition, several studies have utilized α-SMA, vimentin, and FN1 to assess the severity of pulmonary fibrosis during the course of IPF [13,14,15].

Recent gene set enrichment analysis following single-cell RNA sequencing revealed the upregulation of pathways associated with lung fibrosis, including vascular endothelial growth factor (VEGF), wingless gene of *Drosophila* (WNT), transforming growth factor β (TGF-β), and apoptosis [16]. Recently, it has been discovered that treatments targeting the Neu-1 enzyme may be highly effective against SARS-CoV-2 by not only limiting the cytokine storm but also preventing viral entry into cells [17]. Additionally, interleukin-8 may serve as a prognostic indicator for glucocorticoid therapy [18].

Numerous signaling pathways contribute to the pathogenesis of PF, with TGF-β signaling being one of the most extensively studied. The Wnt signaling pathways, particularly the wingless MMTV integration site (Wnt) pathway, have also garnered significant attention for their roles in the exacerbation and progression of chronic lung diseases, especially asthma, chronic obstructive pulmonary disease (COPD), and idiopathic pulmonary fibrosis (IPF) [19]. A proteomic analysis of lung tissue samples from patients who died due to infection with the Alpha variant of SARS-CoV-2 revealed that differentially expressed proteins enriched in the Wnt signaling pathway outnumbered those in the TGF-β pathway [20]. The Wnt signaling pathways include the canonical Wnt/β-catenin signaling pathway and the non-canonical Wnt signaling pathway. Wnt signaling and its effector, β-catenin, have been implicated in the activation of IPF, mediating the chemotaxis and hyperproliferation of alveolar type II epithelial cells [19].

β-catenin, encoded by the CTNNB1 gene (Catenin Beta 1), is a crucial protein in the Wnt/beta-catenin signaling pathway [21]. In the classical Wnt signaling pathway, β-catenin serves as the primary effector responsible for transducing signals to the nucleus. This process initiates the transcription of Wnt-specific genes that play a crucial role in regulating cell fate decisions across various cells and tissues. Excess free β-catenin is phosphorylated by a destruction complex, marking it for subsequent degradation [22]. Aberrant β-catenin expression and its nuclear accumulation have been shown to enhance the transcription of various oncogenes, including *c-Myc* and *cyclin D1*, thereby facilitating tumor initiation, promotion, and progression [23]. Several studies have demonstrated that pulmonary fibrosis, which can result from various causes such as silicosis and idiopathic pulmonary fibrosis, is associated with the accumulation of β-catenin [24,25,26,27]. While previous studies have established that Wnt/β-catenin signaling contributes to PF in various contexts, the specific role of Wnt/β-catenin signaling in SARS-CoV-2-induced fibrosis remains unclear. The aim of this study was to investigate the role of β-catenin in the pathogenesis of PF induced by SARS-CoV-2 infection. Understanding the molecular mechanisms involved can provide a scientific basis for in vitro experiments to identify potential drugs for the treatment of pulmonary fibrosis.

We performed β-catenin knockdown in A549 cells derived from non-small-cell lung cancer that express hACE2. The modified A549-hACE2 cells were then infected with SARS-CoV-2. We conducted molecular biology assays, including immunofluorescence assay (IFA), Western blotting (WB), and quantitative polymerase chain reaction (qPCR) on post-infection cell samples. Additionally, we performed transcriptomics analysis on cellular RNA samples. These approaches validated the mechanism of action and identified other potential effects of β-catenin in SARS-CoV-2-induced PF.

## 2. Materials and Methods

### 2.1. Molecular Cloning

#### 2.1.1. shRNA

shNC

Forward: 5′CCGGTTCTCCGAACGTGTCACGTCTCGAGACGTGACACGTTCGGAGAATTTTTG3′

Reverse: 5′AATTCAAAAATTCTCCGAACGTGTCACGTCTCGAGACGTGACACGTTCGGAGAA3′

shCTNNB1-1

Forward: 5′CCGGCGCATGGAAGAAATAGTTGAACTCGAGTTCAACTATTTCTTCCATGCGTTTTTG3′

Reverse: 5′AATTCAAAAACGCATGGAAGAAATAGTTGAACTCGAGTTCAACTATTTCTTCCATGCG3′

shCTNNB1-2

Forward: 5′CCGGGCTTGGAATGAGACTGCTGATCTCGAGATCAGCAGTCTCATTCCAAGCTTTTTG3′

Reverse: 5′AATTCAAAAAGCTTGGAATGAGACTGCTGATCTCGAGATCAGCAGTCTCATTCCAAGC3′

shCTNNB1-3

Forward: 5′CCGGAGGTGCTATCTGTCTGCTCTACTCGAGTAGAGCAGACAGATAGCACCTTTTTTG3′

Reverse: 5′AATTCAAAAAAGGTGCTATCTGTCTGCTCTACTCGAGTAGAGCAGACAGATAGCACCT3′

#### 2.1.2. Cloning Vectors

For shRNA knockdown of CTNNB1, two primers were annealed and ligated using T4 ligase into the PLKO-NeoR vector that had been cleaved with AgeI and EcoRI. For overexpression of CTNNB1, the coding sequence (CDS) of CTNNB1, as well as a Flag-tagged CTNNB1 CDS, was cloned into the PLVX-NeoR plasmid. An eGFP fluorescent reporter gene plasmid was used as a control.

#### 2.1.3. Cloning Strains

Plasmids were introduced into recipient cells (DH5α) using electroporation and plated on Luria–Bertani (LB) agar. Monoclonal strains were selected every other day and identified through sequencing. The correctly cloned strains were selected for shaker amplification, and the plasmids were extracted for backup in 4 °C.

### 2.2. Cell Culture

All cells used in this study were obtained from the American Type Culture Collection (ATCC). A549-hACE2 cells were cultured in F12K medium supplemented with 10% fetal bovine serum (FBS) (Gibco, Waltham, MA, USA, 21127022), while 293T and Vero cells were cultured in DMEM supplemented with 10% FBS (Gibco, C11995500BT). All cell cultures were maintained at 37 °C in a 5% CO_2_ incubator for passaging.

### 2.3. Cell Line Construction

The constructed plasmids were co-transfected into 293T cells for lentiviral packaging using a triple-plasmid system (plasmid: CG8.91: PMD2.G = 5:3:1). The medium was changed 6 h post-transfection and the supernatant was collected 48 h after the medium change. The supernatant was then centrifuged at 5000 rpm for 10 min and added to pre-plated A549-hACE2 cells for transduction. A concentration of 1.5 mg/mL G418 (Gibco, 10131035) was added for selection after 24 h. Selected cells were harvested once all cells in the negative control wells had died. Cells were gradually expanded in culture and subsequently frozen to establish the modified mixed cell line.

### 2.4. SARS-CoV-2 Infection

SARS-CoV-2 (2019-nCoV-WIV04) used in this study was provided by the National Virus Resource Center (NVRC: IVCAS 6.7512). Vero E6 cells were inoculated with virus for virus amplification. All viral infection experiments were conducted in a Biosafety Level 3 (P3) laboratory at the Wuhan Institute of Virology, Chinese Academy of Sciences.

### 2.5. Sample Testing

#### 2.5.1. IFA

Infected samples were fixed overnight and then permeabilized with 0.2% Triton X-100 (Aladdin, Shanghai, China, T109027) for 10 min. Following permeabilization, samples were washed three times with PBS for 10 min each, blocked with 5% BSA for 1 h, and incubated with the anti-NP antibody (homemade, 1:4000) at 4 °C overnight. Afterward, samples were washed three times with PBS for 10 min each, incubated with a fluorescent secondary antibody at 24 °C for 1 h, and washed again three times with PBS for 10 min each. Finally, nuclei were stained with Hoechst 33258 staining solution for 10 min, followed by three additional washes with PBS for 10 min each. Images were captured using a digital camera under a light microscope (Thermo Fisher Scientific, Waltham, MA, USA, M5000, Evos2 1.5.1402.642-5b07ba).

#### 2.5.2. RT-qPCR

Cell samples were treated with 500 μL of Trizol per well to lyse the cells, and RNA was extracted using the phenol–chloroform method. The lysate was collected at 4 °C. The mixture was then centrifuged at 10,000× *g* for 15 min, and 200 μL of the supernatant was transferred to a new enzyme-free microcentrifuge tube. An equal volume of isopropanol was added, and the sample was mixed gently by inversion. The precipitate was allowed to settle for 10 min at room temperature, followed by centrifugation at 4 °C and 10,000× *g* for 10 min. The supernatant was discarded, and the pellet was washed with 75% ethanol prepared with DEPC-treated water. After centrifugation at 10,000× *g* for 5 min at 4 °C, the supernatant was again discarded, and the pellet was dried until it became clear. Finally, RNA was solubilized using an appropriate volume of DEPC-treated water. A 1 μg sample of RNA was utilized for reverse transcription (Vazyme, Nanjing, China, R323) into complementary DNA (cDNA) using a two-step method. Quantification was performed using a qPCR kit (Vazyme, Q712).

#### 2.5.3. WB

Cells were collected by adding an appropriate amount of lysate, which was then heated at 98 °C in a water bath for 10 min. Afterward, the gel was run using SDS-PAGE. Upon completion, proteins were transferred onto a 0.22 μm PVDF membrane under a constant current of 350 mA in an ice-water bath for 2 h. The membrane was then blocked in 5% skimmed milk powder for 1.5 h. Subsequently, it was incubated overnight with a solution of 0.5% skimmed milk powder diluted with TBST prepared with the appropriate concentration of primary antibody. The following day, the membrane was washed three times with TBST (10 min each time) and then incubated with the secondary antibody at 24 °C for 2 h. The membrane was then washed three more times with TBST (10 min each time) and developed using a color development solution.

The antibodies used in this study were as follows: anti-β-actin (ABclonal, Woburn, MA, USA, AC026, 1:50,000), anti-β-catenin (Proteintech, Rosemont, IL, USA, 51067-2-AP, 1:8000), anti-α-SMA (Proteintech, 14395-1-AP, 1:8000), anti-Vimentin (Proteintech, 10366-1-AP, 1:8000), anti-c-Myc (Proteintech, 10828-1-AP, 1:6000), anti-cyclin D1 (ABclonal, A19038, 1:500), and anti-FN1 (ABclonal, A23830, 1:2000).

### 2.6. Transcriptomics

Total RNA was processed using an mRNA enrichment method. The constructed libraries underwent quality control and, upon passing this assessment, they were sequenced on the DNBSEQ platform with paired-end 150 base pair (PE150) reads. The filtered clean reads were compared against reference sequences and analyzed for Gene Ontology (GO) functional and pathway significance enrichments. RNA sequencing and preliminary analyses were performed using BGI (www.BGI.com).

### 2.7. Statistical Analysis

All experimental procedures were replicated at least three times to ensure the reliability and reproducibility of the results. For the statistical analysis of data derived from two distinct groups, a two-tailed Student’s *t*-test was employed to assess the significance of the differences between them. In the case of multiple group comparisons, a one-way analysis of variance (ANOVA) was conducted to determine whether there were statistically significant differences among the groups. Error bars represent the mean ± SD. A *p*-value < 0.05 was considered statistically significant. Statistical plots were prepared using Prism 9.5.

## 3. Results

### 3.1. Upregulation of β-Catenin and ECM Markers Following SARS-CoV-2 Infection in A549-hACE2 Cells

A549-hACE2 cells were infected with SARS-CoV-2 at a multiplicity of infection (MOI) of 1.0, and samples were collected at 0, 24, 48, and 72 h post-infection. The N protein of SARS-CoV-2 was detected using IFA (Figure 1A). The fluorescence at 0 h was negative, while positive fluorescence was observed at 24, 48, and 72 h, indicating the successful infection of SARS-CoV-2. WB results confirmed that β-catenin, c-Myc, cyclin D1, and α-SMA were upregulated at the protein level. RNA was extracted for RT-qPCR analysis. β-catenin levels were significantly elevated 24 h post-infection. The downstream genes of the β-catenin pathway, *c-Myc* and *cyclin D1*, exhibited an upward trend. Additionally, *α-SMA*, a marker for mesenchymal cells, demonstrated an upregulation trend (Figure 1B,C). The results indicated that ECM components, as well as cellular β-catenin and its downstream signaling molecules, were significantly upregulated after SARS-CoV-2 infection.

### 3.2. Inhibition of β-Catenin Reduces Mesenchymal Accumulation in A549-hACE2 Cells

Knockdown efficiency was assessed following the knockdown of β-catenin, and all three target shRNAs significantly reduced β-catenin levels (Figure 2A). The obtained cells were used to collect samples following SARS-CoV-2 infection. IFA was conducted on fixed cells, and the results indicated that SARS-CoV-2 successfully entered and replicated within the cells (Figure 2B). c-Myc and cyclin D1 levels decreased, and the cellular mesenchymal components α-SMA, vimentin, and FN1 were also downregulated in CTNNB1 knockdown cells. Additionally, transcription of *collagen type III* was suppressed (Figure 2C,D). Expression of c-Myc and cyclin D1, both downstream targets of the β-catenin, was decreased in the presence of β-catenin inhibition and the accumulation of intercellular mesenchymal constituents reduced by SARS-CoV-2 infection.

PLVX-Neo-CTNNB1 (PNC) and PLVX-Neo-CTNNB1-Flag (PNCF) were used for the overexpression of CTNNB1, with the control group consisting of PLVX-Neo-eGFP (eF). The cells successfully overexpressed CTNNB1, and the Flag tag expression was confirmed (Figure 2E). Additionally, the cells were infected with SARS-CoV-2, and the infection efficiency was assessed (Figure 2F). The results indicated that CTNNB1 overexpression did not exacerbate the accumulation of mesenchymal components induced by SARS-CoV-2 infection (Figure 2G,H).

These findings suggest that the knockdown of β-catenin abrogated the upregulation of signaling molecules downstream of β-catenin (*c-Myc, cyclin D1*) and the upregulation of ECM components (*vimentin*, *FN1*, and *COL3A1*) induced by SARS-CoV-2 infection. In contrast, the overexpression of β-catenin did not exacerbate the aforementioned effects caused by SARS-CoV-2 infection.

### 3.3. Transcriptomics of β-Catenin-Modified Cells After SARS-CoV-2 Infection

We used RNA samples collected 48 h after A549-hACE2 cell lines were infected with SARS-CoV-2 to conduct transcriptomic analysis. The negative control (NC) cells comprised uninfected A549-hACE2 cells transfected with a random sequence. We established two comparison groups: the NC+Mock/NC+SARS-2 and NC+SARS-2/shC+SARS-2 group. By identifying the intersection of differentially expressed genes (DEGs) from both comparison groups, we identified 363 DEGs associated with both SARS-CoV-2 infection and *CTNNB1* knockdown. These 363 DEGs were subsequently analyzed using the KEGG pathway and GO analyses. KEGG enrichment analysis showed that the three pathways with the most significant Q values were *Staphylococcus aureus* infection (05150), complement and coagulation cascades (04610), and inflammatory bowel disease (05321). Additionally, the PI3K-Akt signaling pathway (04151), JAK-STAT signaling pathway (04630), and TGF-beta signaling pathway (04350) were significantly enriched (Figure 3A). The KEGG pathways corresponding to these genes were sorted by the number of genes within each pathway, and the ten pathways with the highest gene counts were selected for visualization. The pathways enriched with the highest number of genes were pathways in cancer (05200) with a node connection count of 16, followed by herpes simplex virus 1 infection (05168) and the PI3K-Akt signaling pathway (04151) with node connection counts of 14 and 12, respectively (Figure 3B). The results indicated that the inhibition of β-catenin induced crosstalk among multiple signaling pathways following SARS-CoV-2 infection.

GO enrichment analysis of common genes revealed notable changes at the cellular component level, including the extracellular region (GO:0005576), extracellular space (GO:0005615), collagen-containing extracellular matrix (GO:0062023), and fibrinogen complex (GO:0005577) (Figure 3C). At the cellular process level, the most significant changes included the positive regulation of peptide hormone secretion (GO:0090277), plasminogen activation (GO:0031639), and cellular protein-containing complex assembly. The three genes enriched for plasminogen activation (*FGA*, *FGB*, and *FGG*) were also significantly associated with blood coagulation and fibrin clot formation (GO:0072378) (Figure 3D).

However, after infection with SARS-CoV-2 in the *CTNNB1* overexpression group and the fluorescent reporter control group, the number of DEGs was relatively small, totaling only 45. Among these, 18 genes were upregulated, while 27 were downregulated (Figure 3E). By clustering the expression of these 45 DEGs, we observed that the AKT inhibitor gene *PPP2R2B* was upregulated in the overexpression group (Figure 3F).

## 4. Discussion

The Wnt/β-catenin signaling pathway plays a significant role in PF resulting from various causes, including IPF and paraquat poisoning-induced PF [28,29]. In this study, we observed a significant upregulation of β-catenin, its downstream signaling molecules, and ECM components in A549-hACE2 cells following SARS-CoV-2 infection. Further, the knockdown of β-catenin abrogated the upregulation of both signaling molecules downstream of β-catenin and ECM components induced by SARS-CoV-2 infection. We have preliminarily demonstrated in vitro that β-catenin plays a significant role in pulmonary fibrosis resulting from SARS-CoV-2 infection. Furthermore, our transcriptomic analysis revealed that β-catenin also affects several other disease pathways following SARS-CoV-2 infection.

β-catenin plays a crucial role in various biological processes, including both physiological and pathological mechanisms involved in the development of different organs and associated diseases, such as lung cancer, vascular cancer, bone disorders, neurological conditions, and liver diseases. In lung tissue, the inhibition of the Wnt/β-catenin pathway promotes the rapid and directed differentiation of pluripotent stem cells into proximal airway epithelial cells [21]. In the context of COVID-19, β-catenin has been shown to inhibit macrophage proliferation and repopulation through the β-catenin-HIF-1α axis by disrupting mitochondrial adaptation; however, it can also promote macrophage inflammatory activity [30]. Sanchari et al. demonstrated that β-catenin is essential for effective SARS-CoV-2 infection by utilizing iCRT14, a specific inhibitor of the Wnt/β-catenin signaling pathway [31]. In our study, the Wnt/β-catenin pathway not only directly promotes lung fibrosis induced by SARS-CoV-2 but also has crosstalk with multiple signaling pathways, including the phosphatidylinositol 3-kinase (PI3K)/protein kinase B (PKB/AKT) signaling pathway and Janus kinases (JAK)/ signal transducer and activator of transcription (STAT) signaling pathway.

The PI3K/ AKT signaling pathway is a core signaling pathway that regulates cell growth, proliferation, motility, metabolism, and survival [32]. Evidence suggests that the overexpression of α-SMA in pulmonary fibrosis is associated with activation of the PI3K/AKT pathway [33]. Furthermore, the interaction between TGF-β and PI3K/AKT pathways promotes the development of pulmonary fibrosis [34]. The JAK/STAT signaling pathway mediates a wide range of biological processes, and JAK inhibitors have been approved for use in myelofibrosis; subsequently, the role of JAK/STAT in fibrosis of other organs has come to the attention of researchers [35]. The JAK/STAT pathway is involved in multiple cellular processes in interstitial lung diseases. It promotes the transformation of fibroblasts and epithelial cells into myofibroblasts during fibroblast-to-myofibroblast transition and epithelial-to-mesenchymal transition [36]. Based on previous studies, we hypothesize that the inhibition of β-catenin leads to multi-pathway crosstalk following SARS-CoV-2 infection, which is associated with PF. These complex mechanisms involved warrant further investigation.

We found that the inhibition of β-catenin effectively alleviated PF induced by SARS-CoV-2 infection at the cellular level, suggesting it may serve as a promising drug target for the treatment of COVID-19-related PF. Pirfenidone is an antifibrotic agent with notable anti-inflammatory properties that is utilized in the treatment of fibrotic diseases, including IPF [37]. By analyzing the effects of various concentrations of pirfenidone on the proliferation of the human hepatocellular carcinoma cell line HepG2, Zou et al. demonstrated that pirfenidone can effectively inhibit the Wnt/β-catenin signaling pathway, thereby reducing cell proliferation [38]. This finding suggests that pirfenidone may function as a β-catenin inhibitor, potentially alleviating the symptoms of PF associated with long COVID-19 and thereby reducing the sequelae in patients infected with SARS-CoV-2. Another promising drug is nobiletin, a compound derived from the genus *Chenopodium* and a major constituent of citrus fruits [39]. Nobiletin enhances chemosensitivity to adriamycin and inhibits the Akt/GSK3β/β-catenin/MYCN/MRP1 signaling pathway in A549 cells [40]. Additionally, nobiletin was shown to inhibit the entry of the SARS-CoV-2 pseudovirus into cells by binding to ACE2 and CD147 [41]. Previous studies, consistent with ours, suggest that nobiletin may yield favorable therapeutic outcomes for PF and other symptoms associated with SARS-CoV-2.

However, overexpression of *CTNNB1* did not exacerbate the accumulation of mesenchymal components induced by SARS-CoV-2. Our transcriptomic data indicated that SARS-CoV-2 infection following *CTNNB1* overexpression led to the upregulation of protein phosphatase 2 regulatory subunit Bbeta (PPP2R2B). PPP2R2B functions as an antagonist of the Wnt signaling pathway and inhibits Akt activation [42]. In A549 cells, Akt can upregulate and activate β-catenin, serving as a positive regulator of β-catenin in these cells [43]. Additionally, Akt can influence the regulation of c-Myc as an upstream signaling factor [44]. Therefore, we propose that the overexpression of *CTNNB1* activates PPP2R2B, thereby providing a self-protective mechanism via negative feedback. This process leads to further upregulation of β-catenin, while the baseline expression of β-catenin in A549-hACE2 cells does not exacerbate the accumulation of mesenchymal components. Further studies are needed to fully elucidate this process.

In our analysis of the common DEGs, we found that knockdown of CTNNB1 led to a significant enrichment of genes in the context of *S. aureus* infection, with significance observed in the comparison group infected with SARS-CoV-2. Comprehensive transcriptomic analysis, bioinformatics assessments, and critical aspects of experimental validation of *S. aureus* infection are specifically characterized by the activation of the Wnt/β-catenin signaling pathway, which leads to inhibition of NF-κB signaling activity [45]. Therefore, SARS-CoV-2 and *S. aureus* infection in human lung-derived cells potentially share a similar mechanism of action involving the β-catenin signaling pathway.

Another important pathway identified is the viral myocarditis pathway (05416). Similarly, common DEGs were enriched in the knockdown and SARS-CoV-2-infected comparison groups. Viral myocarditis has been observed in COVID-19 patients since the onset of the SARS-CoV-2 pandemic, with mortality from cardiovascular causes higher in these patients compared to those infected with other viruses [46]. Sophie et al. analyzed previous case reports and concluded that the presence of the virus in the hearts of patients with COVID-19 is rare. They suggested that the clinical manifestations of fulminant myocarditis are caused by indirect mechanisms, such as cytokine storms, rather than direct viral effects mediated by SARS-CoV-2 [47]. Coxsackievirus B3, a known cause of viral myocarditis, has been shown to facilitate crosstalk between the ERK1/2 signaling pathway and the Wnt/β-catenin pathway, promoting viral replication and pathogenicity in host cells [48]. Our data proposed that *CTNNB1* knockdown can lead to activation of the viral myocarditis pathway following SARS-CoV-2 infection. However, whether this effect is a direct or indirect action, as well as the specific mechanisms involved, requires further investigation.

A limitation of our study is that it was conducted solely at the cellular level without including animal studies. In addition, this study utilized only A549 cells that are derived from human non-small-cell lung carcinoma and have been employed in several investigations of pulmonary fibrosis. A more comprehensive validation of the association between β-catenin and SARS-CoV-2-associated pulmonary fibrosis would require the inclusion of healthy human alveolar epithelial cell lines in addition to lung cancer cell lines.

Pulmonary fibrosis induced by SARS-CoV-2 infection represents a complex pathological process that involves the interaction of various cell types and multiple signaling pathways. Future studies should consider integrating β-catenin with other known signaling pathways associated with pulmonary fibrosis to comprehensively evaluate their synergistic effects and enhance our understanding of the pathogenesis of this condition. β-catenin is anticipated to be a potential drug target for the treatment of SARS-CoV-2-related pulmonary fibrosis, as the inhibition of β-catenin has been shown to reduce the accumulation of ECM resulting from SARS-CoV-2 invasion. Furthermore, as our understanding of the regulatory mechanisms of β-catenin deepens, a combination of drugs that targets both the upstream and downstream signaling pathways of β-catenin may achieve more effective therapeutic outcomes, thereby expanding the options for clinical treatment strategies.

## 5. Conclusions

In conclusion, we successfully constructed a model of cellular fibrosis induced by SARS-CoV-2 (2019-nCoV-WIV04) infection in a human lung-derived cell line (A549-hACE2). It confirms that β-catenin, a key component of the Wnt/β-catenin signaling pathway, plays a significant role in lung fibrosis induced by SARS-CoV-2 infection. Additionally, the inhibition of β-catenin mitigates the mesenchymal accumulation resulting from the invasion of A549-hACE2 cells by SARS-CoV-2. Bioinformatics analyses further indicated that β-catenin knockdown also resulted in crosstalk in the PI3K-Akt, JAK-STAT, and TGF-β signaling pathways. The increased expression of PPP2R2B following β-catenin overexpression may play a role in protecting cells from exacerbating fibrosis; however, the precise mechanism requires further investigation.

## Figures and Tables

**Figure 1 cells-14-00394-f001:**
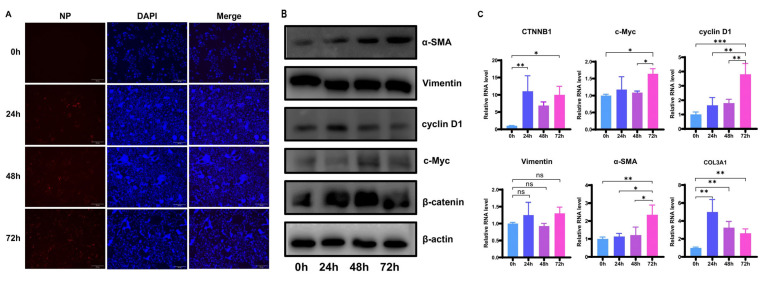
Significant upregulation of β-catenin, its downstream signaling molecules, and ECM components in A549-hACE2 cells following SARS-CoV-2 infection. (**A**) IFA of N Protein of SARS-CoV-2 to detect efficiency of infection at various time points. 200× magnification for all fields of view. (**B**) WB image of β-catenin, c-Myc, cyclin D1, α-SMA, and Vimentin following A549-hACE2 infection with SARS-CoV-2. (**C**) qPCR of β-catenin, c-Myc, cyclin D1, α-SMA, Vimentin, and COL3A1 following A549-hACE2 infection with SARS-CoV-2. A *p*-value < 0.05 (denoted by “*”) was considered statistically significant. Differences with *p*-value < 0.01 are denoted by “**”, while differences with *p*-value < 0.005 are denoted by “***”. Non-significant differences are denoted by “ns”.

**Figure 2 cells-14-00394-f002:**
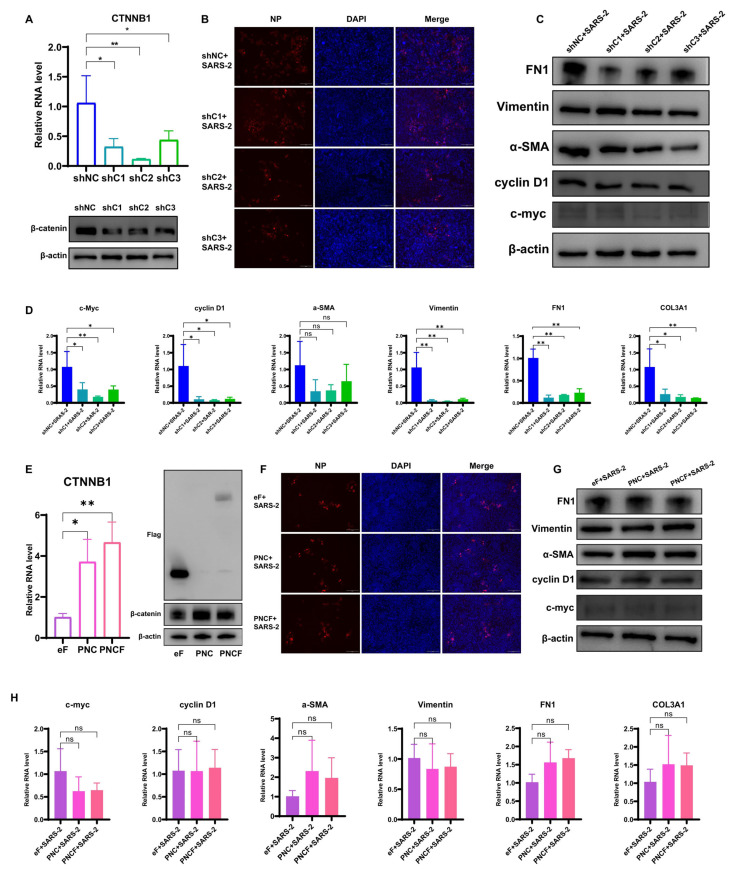
Inhibition of β-catenin alleviated fibrosis following SARS-CoV-2 infection in A549-hACE2 cells, while overexpression of β-catenin did not exacerbate fibrosis. (**A**) Knockdown efficiency test of *CTNNB1.* (**B**) Detection efficiency of SARS-CoV-2 infection. 200× magnification for all fields of view. (**C**) WB image of c-Myc, cyclin D1, α-SMA, Vimentin, and FN1 following A549-hACE2-shCTNNB1 infection with SARS-CoV-2. (**D**) qPCR of *c-myc*, *cyclin D1*, *a-SMA*, *Vimentin*, *COL3A1*, and *FN1* following A549-shACE2 infection with SARS-CoV-2. (**E**) Overexpression efficiency test of *CTNNB1.* (**F**) Detection efficiency of SARS-CoV-2 infection. 200× magnification for all fields of view. (**G**) WB image of c-Myc, cyclin D1, α-SMA, Vimentin, and FN1 following A549-hACE2-CTNNB1 infection with SARS-CoV-2. (**H**) qPCR of *c-myc*, *cyclin D1*, *a-SMA*, *Vimentin*, *COL3A1*, and *FN1* following A549-ACE2 infection with SARS-CoV-2. A *p*-value < 0.05 (denoted by “*”) was considered statistically significant. Differences with *p*-value < 0.01 are denoted by “**”. Non-significant differences are denoted by “ns”.

**Figure 3 cells-14-00394-f003:**
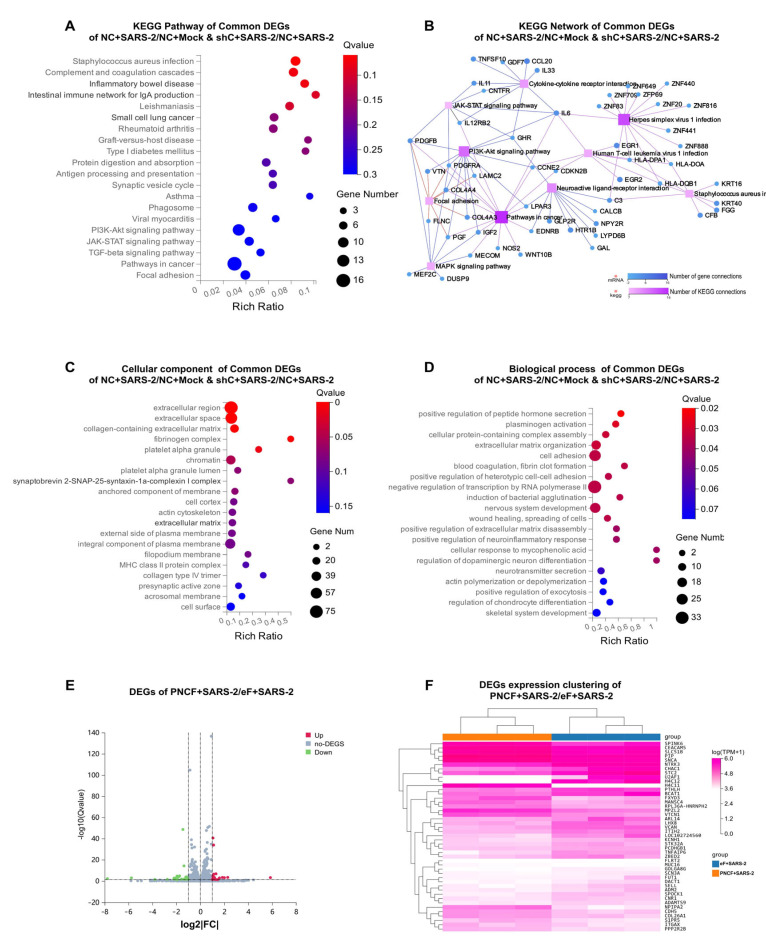
(**A,B**) KEGG pathway and network of common DEGs of NC+SARS-2/NC+Mock and shC+SARS-2/NC+SARS-2. Inhibition of β-catenin induces crosstalk among multiple signaling pathways following SARS-CoV-2 infection. (**C**,**D**) Cellular component and biological process of common DEGs of NC+SARS-2/NC+Mock and shC+SARS-2/NC+SARS-2. GO enrichment analysis confirmed that the inhibition of β-catenin altered components, including collagen-containing ECM components and fibrinogen complexes, in A549-hACE2 cells following SARS-CoV-2 infection. (**E**,**F**) Volcanic plot and expression clustering of DEGs of eF+ SARS-2/PNCF+ SARS-2. Clustering and visualization of 45 DEGs revealed that PPP2R2B was upregulated in the CTNNB1 overexpression group.

## Data Availability

The data that support the findings of this study will be available in the China National GeneBank DataBase at DOI:10.26036/CNP0006698.

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
