# Peer review of "Mechanism of β-Catenin in Pulmonary Fibrosis Following SARS-CoV-2 Infection"

_cells, 2025, doi:10.3390/cells14060394_

Round 1
Reviewer 1 Report
Comments and Suggestions for Authors
It's not clear how this study is related to pulmonary fibrosis.
Major comments:
1. β-catenin usually overexpressed in both acute and chronic lung injury and could not be a marker for IPF;
2. The chosen cell line is cancer cells. To study fibrosis the authors should use macrophages to study profibrotic pathways;
3. No analyses on traditional profibrotic pathways like TGFb, SMAD3 activation, ERK activation, HSP90, or Collagen Type I were presented.Coll3A
Author Response
Reviewer #1:
It's not clear how this study is related to pulmonary fibrosis.
Major comments:
Comments1: β-catenin usually overexpressed in both acute and chronic lung injury and could not be a marker for IPF;
Response1: β-catenin, encoded by the CTNNB1 gene (Catenin Beta 1), is a crucial protein in the Wnt/beta-catenin signaling pathway. In the classical Wnt signaling pathway, β-catenin serves as the primary effector responsible for transducing signals to the nucleus. This process initiates the transcription of Wnt-specific genes that play a crucial role in regulating cell fate decisions across various cells and tissues. Excess free β-catenin is phosphorylated by a destruction complex, marking it for subsequent degradation. Aberrant β-catenin expression and its nuclear accumulation have been shown to enhance the transcription of various oncogenes, including c-Myc and cyclin D1, thereby facilitating tumor initiation, promotion, and progression. Several studies have demonstrated that pulmonary fibrosis, which can result from various causes such as silicosis and idiopathic pulmonary fibrosis, is associated with the accumulation of β-catenin.
The aforementioned has been incorporated into the manuscript. (L79–89)
Comments2: The chosen cell line is cancer cells. To study fibrosis the authors should use macrophages to study profibrotic pathways;
Response2: We chose A549 cells based on previous studies of PF as many researchers opted to use these parenchymal cells derived from non-small cell lung carcinoma. If sufficient resources and conditions are available, other cells should be utilized to comprehensively demonstrate the association between β-catenin and SARS-CoV-2-associated pulmonary fibrosis. We have included this consideration in the limitations section of the manuscript. (L420–424)
Comments3: No analyses on traditional profibrotic pathways like TGFb, SMAD3 activation, ERK activation, HSP90, or Collagen Type I were presented. Col13A
Response3: The ECM consists of a variety of proteins, among which fibronectin (FN) is a significant glycoprotein. FN is one of the first ECM proteins to be assembled during the initial stages of tissue development and wound healing. Type III collagen (COL3A1), an ECM protein, is synthesized by cells as a pre-procollagen. α-smooth muscle actin (α-SMA) is a well-characterized protein that serves as a marker for activated fibroblasts in various tissues and organs, including the lung. Vimentin, a member of the intermediate filament family of proteins, is essential for epithelial-mesenchymal transition. In addition, several studies have utilized α-SMA, vimentin, and FN1 to assess the severity of pulmonary fibrosis during the course of IPF.
We choses to use these markers of pulmonary fibrosis. Of course, future research could focus on the markers you mentioned, which would necessitate further investigation. The aforementioned text has been incorporated into the manuscript. (L50–59)
Reviewer 2 Report
Comments and Suggestions for Authors
The topic is interesting and the paper is quite well written. I have some comments:
1) Abstract. L 13-17. I suggest putting the most important information in the results to attract the attention of potential readers.
2) Abstract. L 18-22. Abstract might be beneficial to include a sentence that briefly summarizes the key findings of the study. This can provide readers with a quick overview of the research.
3) Introduction. L 37-42. During the initiation phase of PF, stress and immune responses trigger the activation of multiple pro-inflammatory pathways. This is followed by a proliferative phase in which fibroblasts undergo differentiation and proliferation. Authors are kindly requested to emphasize the current concepts about these issues in the context of recent knowledge and the available literature. These articles should be quoted in the References list. References 1. Novel Therapeutic Target Critical for SARS-CoV-2 Infectivity and Induction of the Cytokine Release Syndrome. Cells. 2023;12(9):1332. Published 2023 May 7. doi:10.3390/cells12091332
2. Cytokine Profiles as Potential Prognostic and Therapeutic Markers in SARS-CoV-2-Induced ARDS. J Clin Med. 2022;11(11):2951. Published 2022 May 24. doi:10.3390/jcm11112951
4) Introduction. L 67-76. I suggest to underline the aim of the study.
5) I suggest using acronyms only when strictly necessary to facilitate reading and attract readers.
6) 2.7. Statistical analysis 187 All experiments were repeated at least three times. Data from two groups were ana-188 lyzed using two-tailed t-tests, and multiple group comparisons were analyzed using one-189 way ANOVA. Error bars represent the mean ± SD. A p-value < 0.05 (denoted by “*”) was 190 considered statistically significant. Differences with p-value < 0.01 are denoted by “**”, 191 while differences with p-value < 0.005 are denoted by “***”. Statistical plots were prepared 192 using Prism 9.5. I suggest to improve the description of statistical tests used to evaluate the data.
7) 3. Results 194-285.The paragraph is quite rumbling and difficult to read. I suggest to underline the most important results to support the conclusions.
8) 4. Discussion 286 The Wnt/β-catenin signaling pathway plays a significant role in PF resulting from 287 various causes, including IPF, paraquat poisoning-induced PF, and COPD [13,14]. In this 288 study, we confirmed that β-catenin is a key factor in the pathogenesis of lung fibrosis in-289 duced by SARS-CoV-2 infection at the cellular level. Furthermore, our transcriptomic 290 analysis revealed that β-catenin also affects several other disease pathways following 291 SARS-CoV-2 infection. The discussion section needs to be improved. I suggest to underline the limitations of the study. It is necessary to be more concise in the presentation of the facts, clarifying the results obtained and comparing them with previous published literature.
9) In conclusion, this study confirms that β-catenin, a key component of the Wnt/β-362 catenin signaling pathway, plays a significant role in lung fibrosis induced by SARS-CoV-363 2 infection. Additionally, the inhibition of β-catenin mitigates the mesenchymal accumu-364 lation resulting from the invasion of A549-hACE2 cells by SARS-CoV-2. Bioinformatics 365 analyses further indicated that β-catenin knockdown also resulted in crosstalk in the PI3K-366 Akt and JAK-STAT signaling pathways. I suggest to underline the future prospects
Author Response
Reviewer #2:
The topic is interesting and the paper is quite well written. I have some comments:
Comments1: Abstract. L 13-17. I suggest putting the most important information in the results to attract the attention of potential readers.
Response1: We have revised the abstract to include the most important information and conclusions of this study. (L8–24)
Comments2: Abstract. L 18-22. Abstract might be beneficial to include a sentence that briefly summarizes the of the study. This can provide readers with a quick overview of the research.
Response2: We briefly summarize our key findings as follows: The results of our mechanistic studies indicate that β-catenin plays a significant role in lung fibrosis resulting from SARS-CoV-2 infection. Furthermore, the inhibition of β-catenin mitigated the accumulation of mesenchymal stroma in A549-hACE2 cells. Additionally, β-catenin knockdown was found to facilitate multi-pathway crosstalk following SARS-CoV-2 infection.β-catenin overexpression did not exacerbate cellular fibrosis may be attributed to the activation of PPP2R2B.
We included the aforementioned information in the abstract. (L20–24)
Comments3: Introduction. L 37-42. During the initiation phase of PF, stress and immune responses trigger the activation of multiple pro-inflammatory pathways. This is followed by a proliferative phase in which fibroblasts undergo differentiation and proliferation. Authors are kindly requested to emphasize the current concepts about these issues in the context of recent knowledge and the available literature. These articles should be quoted in the References list. References 1. Novel Therapeutic Target Critical for SARS-CoV-2 Infectivity and Induction of the Cytokine Release Syndrome. Cells. 2023;12(9):1332. Published 2023 May 7. doi:10.3390/cells12091332
- Cytokine Profiles as Potential Prognostic and Therapeutic Markers in SARS-CoV-2-Induced ARDS. J Clin Med. 2022;11(11):2951. Published 2022 May 24. doi:10.3390/jcm11112951
Response3: We have provided a more detailed explanation of the development process of PF (L43–46) and included information on our selection of ECM markers (L50–59). Thank you for the references; we have incorporated them into the existing literature (L63–66).
Comments4: Introduction. L 67-76. I suggest to underline the aim of the study.
Response4: The aim of this study was to investigate the role of β-catenin in the pathogenesis of PF induced by SARS-CoV-2 infection. Understanding the molecular mechanisms involved can provide a scientific basis for in vitro experiments to identify potential drugs for the treatment of pulmonary fibrosis.
This was added to L92–95 of the manuscript.
Comments5: I suggest using acronyms only when strictly necessary to facilitate reading and attract readers.
Response5: We have thoroughly examined the manuscript and made several revisions. In lines 50–59, we have clarified the abbreviation for the ECM markers, and in lines 79–80, we have described CTNNB1. Additionally, in line 285, we have defined DEGs, and in lines 373-381, we have removed the unnecessary abbreviations for pirfenidone and nobiletin.
Comments6: 2.7. Statistical analysis 187 All experiments were repeated at least three times. Data from two groups were ana-188 lyzed using two-tailed t-tests, and multiple group comparisons were analyzed using one-189 way ANOVA. Error bars represent the mean ± SD. A p-value < 0.05 (denoted by “*”) was 190 considered statistically significant. Differences with p-value < 0.01 are denoted by “**”, 191 while differences with p-value < 0.005 are denoted by “***”. Statistical plots were prepared 192 using Prism 9.5. I suggest to improve the description of statistical tests used to evaluate the data.
Response6: We have modified the description of the statistics and clarified the statistical methods (L213–218). In addition, we have removed statements that duplicate the content of the figure notes (Differences with p-value < 0.01 are denoted by “**”, while differences with p-value < 0.005 are denoted by “***”).
Comments7: 3. Results 194-285.The paragraph is quite rumbling and difficult to read. I suggest to underline the most important results to support the conclusions.
Response7: To enhance the organization of the results, we have included a brief summary following each section, which is underlined. (L232–234, 262–266, 298–299, 324–325)
Comments8: 4. Discussion 286 The Wnt/β-catenin signaling pathway plays a significant role in PF resulting from 287 various causes, including IPF, paraquat poisoning-induced PF, and COPD [13,14]. In this 288 study, we confirmed that β-catenin is a key factor in the pathogenesis of lung fibrosis in-289 duced by SARS-CoV-2 infection at the cellular level. Furthermore, our transcriptomic 290 analysis revealed that β-catenin also affects several other disease pathways following 291 SARS-CoV-2 infection. The discussion section needs to be improved. I suggest to underline the limitations of the study. It is necessary to be more concise in the presentation of the facts, clarifying the results obtained and comparing them with previous published literature.
Response8: In lines 328–334, we summarize our conclusions for clarity. In lines 352–365, we relate the transcriptomic results to previously published literature to provide a more comprehensive explanation and prediction of our findings. Additionally, in lines 419–424, we discuss the limitations of our study.
Comments9: In conclusion, this study confirms that β-catenin, a key component of the Wnt/β-362 catenin signaling pathway, plays a significant role in lung fibrosis induced by SARS-CoV-363 2 infection. Additionally, the inhibition of β-catenin mitigates the mesenchymal accumu-364 lation resulting from the invasion of A549-hACE2 cells by SARS-CoV-2. Bioinformatics 365 analyses further indicated that β-catenin knockdown also resulted in crosstalk in the PI3K-366 Akt and JAK-STAT signaling pathways. I suggest to underline the future prospects
Response9: Pulmonary fibrosis induced by SARS-CoV-2 infection represents a complex pathological process that involves the interaction of various cell types and multiple signaling pathways. Future studies should consider integrating β-catenin with other known signaling pathways associated with pulmonary fibrosis to comprehensively evaluate their synergistic effects and enhance our understanding of the pathogenesis of this condition. β-catenin is anticipated to be a potential drug target for the treatment of SARS-CoV-2-related pulmonary fibrosis, as the inhibition of β-catenin has been shown to reduce the accumulation of ECM resulting from SARS-CoV-2 invasion. Furthermore, as our understanding of the regulatory mechanisms of β-catenin deepens, a combination of drugs that targets both the upstream and downstream signaling pathways of β-catenin may achieve more effective therapeutic outcomes, thereby expanding the options for clinical treatment strategies.
The aforementioned has been incorporated into the manuscript. (L425–436)
Reviewer 3 Report
Comments and Suggestions for Authors
1. The authors have undertaken a sophisticated molecular/cellular study to investigate the Wnt/β-catenin pre-EMT system in commercial cultured small cell lung cancer cloned cells primed for the ability to accept the COVID-virus. As far as I can tell, the laboratory work has been very carefully and expertly done, though laboratory molecular work is not my foremost expertise. The results show that in this cell system COVID-virus infection does activate this system quite comprehensively. This is interesting and relevant work which may be significant to broader situations of pulmonary fibrosis, but there is a tendency to over generalise from a very focused cell system. There are some issues which need dealing with.
2. The authors have used an airway epithelial cancer cell, rather than a modified type 2 alveolar epithelial cell, which might be more relevant to lung parenchyma fibrotic pathology. Perhaps the authors could defend that in the first instance. Later they should not make claims that they have “proved” that the Wnt/β-catenin system is important in driving pulmonary fibrosis. However, they have actually undertaken an in-vitro lung cancer cell system study and are now in a position to hypothesise this, although there is already preliminary evidence from fibrotic lung analysis that this might be the case. This is not meant to undermine the quality of this study but just to try and give it some degree of proportion.
3. The writing in general might be taken as rather “jargonish”, and if this is to be useful for a more general scientific audience in the respiratory field I think some technical touching up/rewriting is required. In particular, there is an almost overwhelming use of abbreviations which are not introduced, even in subtitles, and this goes considerably beyond standard laboratory items; good examples might be CTNNB1 and DEGs/KEGs.
4. The Abstract could be simplified somewhat to be even more focused. The concepts are really pretty straightforward as is the result. There is no need to mention the ACE-receptor, for example, this is merely a tool used for cellular infection; but if it is felt necessary to introduce it then his purpose and significance needs to be spelt out.
5. Most of the studies a nicely presented and are easy to follow, but there's a great deal of information in the figures, some of which is not spelled out either in the text as such or even in the legends and the authors need to be a bit more careful about this. A particular example is COL3A1 in figure 1A, but there are others.
6. I am less happy about the Transcriptomic final study, partly because I'm not totally sure of what the over- or under-expression of β-catenin means without some data on its cytoplasmic on nuclear levels in the cell, or that of other components of the Wnt system downstream. Having said that, seeing these other molecular systems that come up in the study is tantalising. The authors need to say more about the relevance of some of these to the core issue of fibrosis. The only one that seems obvious is the TGFbeta data; that makes me wish even more that that system had been investigated in the core studies as well as the Wnt/β-catenin one. What relevance might the P13K and JAK data have, specially to fibrosis?
7. Lesser points:
A. The very opening sentence of the Abstract is rather over-catastrophising! I think something a bit more straightforward, such as: “Significant morbidity and mortality can result from SARS-CoV-2 lung infection...”
B. Ideally in line 15 the word “stably” should be moved to avoid a grammatically awkward split infinitive.
C. Since you have raised the issue of epithelial cell surface hACE-2, it may be worth saying something about its normal expression in the lung and the situations in which this may be elevated, presumably making individuals prone to more serious infection. Could this be related to inflammation? You mentioned inflammation a couple of times, and it would be good to give it some relevance!
Comments on the Quality of English LanguageI have made some comments for improvement ... but overall vey good.
Author Response
Reviewer #3:
Comments1: The authors have undertaken a sophisticated molecular/cellular study to investigate the Wnt/β-catenin pre-EMT system in commercial cultured small cell lung cancer cloned cells primed for the ability to accept the COVID-virus. As far as I can tell, the laboratory work has been very carefully and expertly done, though laboratory molecular work is not my foremost expertise. The results show that in this cell system COVID-virus infection does activate this system quite comprehensively. This is interesting and relevant work which may be significant to broader situations of pulmonary fibrosis, but there is a tendency to over generalise from a very focused cell system. There are some issues which need dealing with.
Response1: We have improved the expression and clarity of the article, and we hope you will be satisfied with the revisions.
Comments2: The authors have used an airway epithelial cancer cell, rather than a modified type 2 alveolar epithelial cell, which might be more relevant to lung parenchyma fibrotic pathology. Perhaps the authors could defend that in the first instance. Later they should not make claims that they have “proved” that the Wnt/β-catenin system is important in driving pulmonary fibrosis. However, they have actually undertaken an in-vitro lung cancer cell system study and are now in a position to hypothesise this, although there is already preliminary evidence from fibrotic lung analysis that this might be the case. This is not meant to undermine the quality of this study but just to try and give it some degree of proportion.
Response2: We agree that modified type 2 alveolar epithelial cells are more suitable for use as a cell line in studying PF than A549 cells. However, sourcing and culturing modified type 2 alveolar epithelial cells presents significant challenges. Our choice of A549 cells is based on previous studies of PF, and many researchers opt to use these parenchymal cells derived from non-small cell lung carcinoma. If sufficient resources and conditions are available, modified type 2 alveolar epithelial cells should be utilized to comprehensively demonstrate the association between β-catenin and SARS-CoV-2-associated pulmonary fibrosis. We have included this consideration in the limitations section of the manuscript. (L419–424)
We have preliminarily demonstrated in vitro that β-catenin plays a significant role in pulmonary fibrosis resulting from SARS-CoV-2 infection. (L332–334)
Comments3: The writing in general might be taken as rather “jargonish”, and if this is to be useful for a more general scientific audience in the respiratory field I think some technical touching up/rewriting is required. In particular, there is an almost overwhelming use of abbreviations which are not introduced, even in subtitles, and this goes considerably beyond standard laboratory items; good examples might be CTNNB1 and DEGs/KEGs.
Response3: We have amended the abbreviations accordingly. We have thoroughly examined the manuscript and made several revisions. In lines 50–59, we have clarified the abbreviations for the ECM markers, and in lines 79–80, we have described CTNNB1. Additionally, in line 285, we have defined DEGs, and in lines 373-381, we have removed the unnecessary abbreviations for pirfenidone and nobiletin.
Comments4: The Abstract could be simplified somewhat to be even more focused. The concepts are really pretty straightforward as is the result. There is no need to mention the ACE-receptor, for example, this is merely a tool used for cellular infection; but if it is felt necessary to introduce it then his purpose and significance needs to be spelt out.
Response4: We have revised the abstract to retain the essential information and provide a concise summary of the study's results. (L8–24)
Comments5: Most of the studies a nicely presented and are easy to follow, but there's a great deal of information in the figures, some of which is not spelled out either in the text as such or even in the legends and the authors need to be a bit more careful about this. A particular example is COL3A1 in figure 1A, but there are others.
Response5: We have modified all figure legends to retain essential descriptions, eliminate repetitive information, and summarize the experimental results presented in each figure as needed. (L235–243, L269–270, L304–312)
Additionally, we have summarized the experimental results in the main text for readers' convenience. (L232–234, 262–266, 298–299, 324–325)
Comments6: I am less happy about the Transcriptomic final study, partly because I'm not totally sure of what the over- or under-expression of β-catenin means without some data on its cytoplasmic on nuclear levels in the cell, or that of other components of the Wnt system downstream. Having said that, seeing these other molecular systems that come up in the study is tantalising. The authors need to say more about the relevance of some of these to the core issue of fibrosis. The only one that seems obvious is the TGFbeta data; that makes me wish even more that that system had been investigated in the core studies as well as the Wnt/β-catenin one. What relevance might the P13K and JAK data have, specially to fibrosis?
Response6: The PI3K/ AKT signaling pathway is a core signaling pathway that regulates cell growth, proliferation, motility, metabolism and survival. Evidence suggests that the overexpression of α-SMA in pulmonary fibrosis is associated with activation of the PI3K/AKT pathway. Furthermore, the interaction between TGF-β and PI3K/AKT pathways promotes the development of pulmonary fibrosis. The JAK/STAT signaling pathway mediates a wide range of biological processes, and JAK inhibitors have been approved for use in myelofibrosis; subsequently, the role of JAK/STAT in fibrosis of other organs has come to the attention of researchers. The JAK/STAT pathway is involved in multiple cellular processes in interstitial lung diseases. It promotes the transformation of fibroblasts and epithelial cells into myofibroblasts during fibroblast-to-myofibroblast transition and epithelial-to-mesenchymal transition. Based on previous studies, we hypothesize that the inhibition of β-catenin leads to multi-pathway crosstalk following SARS-CoV-2 infection, which is associated with PF. These complex mechanisms involved warrant further investigation.
We have incorporated the aforementioned text into the discussion section of the manuscript. (L352–365)
Comments7: Lesser points:
Comments A: The very opening sentence of the Abstract is rather over-catastrophising! I think something a bit more straightforward, such as: “Significant morbidity and mortality can result from SARS-CoV-2 lung infection...”
Response A: We have revised the summary section to enhance its conciseness.
Comments B: Ideally in line 15 the word “stably” should be moved to avoid a grammatically awkward split infinitive.
Response B: We have revised the abstract accordingly.
Comments C: Since you have raised the issue of epithelial cell surface hACE-2, it may be worth saying something about its normal expression in the lung and the situations in which this may be elevated, presumably making individuals prone to more serious infection. Could this be related to inflammation? You mentioned inflammation a couple of times, and it would be good to give it some relevance!
Response C: ACE2 is predominantly located on the apical surface of well-differentiated airway epithelial cells, particularly in ciliated cells. Since viruses must attach to and enter cells before replication, the surface localization of ACE2 and the state of cellular differentiation may significantly influence SARS-CoV-2 disease and other pathologies associated with acute lung injury (Pulmonary Angiotensin-Converting Enzyme 2 (ACE2) and Inflammatory Lung Disease. SHOCK 46(3):p 239-248, September 2016. | DOI: 10.1097/SHK.0000000000000633). In a study of autopsy samples, ACE2 protein expression was found to be significantly lower in the lung tissue of normal controls compared to that of patients with COVID-19 (ACE2 protein expression in lung tissues of severe COVID-19 infection. Scientific reports, 12(1), 4058.). In our study, hACE2 was introduced to A549 cells as a receptor necessary for the invasion of these cells by SARS-CoV-2.
Most cases of PF arise from previous acute lung inflammation. If these inflammatory reactions are not resolved promptly, they can result in the deposition of fibrous tissue in the lungs, which may subsequently lead to the development of PF. Therefore, inflammation is, in fact, an early manifestation of PF. These points have been added to the manuscript.
Round 2
Reviewer 2 Report
Comments and Suggestions for Authors
The authors adequately answered my questions. I have no further comments.